# Choice of Polymer, but Not Mesh Structure Variation, Reduces the Risk of Bacterial Infection with *Staphylococcus aureus* In Vivo

**DOI:** 10.3390/biomedicines11072083

**Published:** 2023-07-24

**Authors:** Sophia M. Schmitz, Marius J. Helmedag, Andreas Kroh, Daniel Heise, Uwe Klinge, Andreas Lambertz, Mathias W. Hornef, Ulf P. Neumann, Roman M. Eickhoff

**Affiliations:** 1Department of General, Visceral and Transplantation Surgery, RWTH Aachen University Hospital, Pauwelsstr. 30, 52074 Aachen, Germany; 2Department of Medical Microbiology, RWTH Aachen University, Pauwelsstr. 30, 52074 Aachen, Germany

**Keywords:** mesh infection, hernia, foreign body reaction, mesh material, mesh structure

## Abstract

Background: Synthetic mesh material is of great importance for surgical incisional hernia repair. The physical and biochemical characteristics of the mesh influence mechanical stability and the foreign body tissue reaction. The influence on bacterial infections, however, remains ill-defined. The aim of the present study was to evaluate the influence of a modified mesh structure with variation in filament linking on the occurrence of bacterial infection that is indicated by the occurrence of CD68^+^, CD4^+^, and CD8^+^ cells in two different materials. Methods: A total of 56 male Sprague Dawley rats received a surgical mesh implant in a subcutaneous abdominal position. The mesh of two different polymers (polypropylene (PP) and polyvinylidenfluoride (PVDF)) and two different structures (standard structure and bold structure with higher filament linking) were compared. During the implantation, the meshes were infected with *Staphylococcus (S.) aureus*. After 7 and 21 days, meshes were explanted, and the early and late tissue responses to infection were histologically evaluated. Results: Overall, the inflammatory tissue response was higher at 7 days when compared to 21 days. At 7 days, PP meshes of the standard structure (PP-S) showed the strongest inflammatory tissue response in comparison to all the other groups. At 21 days, no statistically significant difference between different meshes was detected. CD8^+^ cytotoxic T cells showed a significant difference at 21 days but not at 7 days. PP meshes of both structures showed a higher infiltration of CD8^+^ T cells than PVDF meshes. CD4^+^ T helper cells differed at 7 days but not at 21 days, and PVDF meshes in a bold structure showed the highest CD4^+^ T cell count. The number of CD68^+^ macrophages was also significantly higher in PP meshes in a standard structure when compared to PVDF meshes at 21 days. Conclusion: The inflammatory tissue response to *S. aureus* infection appears to be highest during the early period after mesh implantation. PP meshes showed a higher inflammatory response than PVDF meshes. The mesh material appears to be more important for the risk of infection than the variation in filament linking.

## 1. Introduction

Synthetic meshes are widely used as the prosthetic graft material for hernia repair. The ideal mesh is the subject of continuous discussion, as meshes for hernia repair have to provide high mechanical stability and biocompatibility. However, little is known about the influence of the mesh material and structure on the risk of bacterial infections and the subsequent inflammatory tissue response [1,2].

While patient-related factors that are associated with mesh infection, such as age and comorbidities, cannot be influenced, an optimized mesh structure could potentially lower the risk of infection and reduce an adverse tissue reaction.

A meta-analysis by Mavros et al. identified smoking, an American Society of Anesthesiologists (ASA) score ≥ 3, a patient’s age, and the duration and emergency setting of the operation as independent risk factors for infection [3].

Polypropylene (PP) is used most commonly as a permanent mesh material [4]. Other materials include polyvinylidenfluoride (PVDF), expanded polytetrafluoroethylene (ePTFE), and polyester-based meshes and composites [4,5]. Published data suggest that PP meshes might be less prone to bacterial adhesion than PE and microporous ePTFE materials [6,7]. In terms of mesh structure, light-weight macroporous meshes have been shown to be more resistant to bacterial adhesion compared to high-weight microporous meshes in vitro and in vivo [8,9,10]. Also, PVDF meshes appear to be linked to a lower inflammatory tissue response than PP meshes [1,11]. PVDF meshes have even been used for abdominal wall reconstruction in cases of enteric fistulas in a setting of massive bacterial exposure with favorable results [12]. Engelsman et al. showed an increased risk for mesh infection with a larger mesh surface, either due to the size of the material used or the filamentous structure [9]. Aydinuraz et al. reported an influence of the mesh composition on bacterial adherence [13]. The superiority of large-pore meshes might be due to decreased scar formation, or “bridging” fibrosis, that has been reported to influence biocompatibility [14]. In multifilament meshes, the increased surface area may promote the adherence of bacteria [15]. Thus, mesh material, size, and structure may be linked to higher risks of bacterial infection [1].

The influence of the surface mesh characteristics has been investigated with respect to the mesh pore size. Through variations in filament linking, the structure of a mesh can be altered without changing the filament length or surface. Thus, the advantages of large-pore meshes might be combined with a possible mechanical superiority. However, structural differences with respect to filament linking or ultrastructure have not been the subject of an investigation.

Here, monofilament PP and PVDF meshes of two different structures (a standard and a modified, bold structure with higher filament linking) were compared. The standard PP mesh served as a benchmark to compare our results with the results from other studies. The aim of this study was to assess the influence of a mesh structure with equal surface area on the risk of bacterial infection in a rat model in vivo.

## 2. Material and Methods

### 2.1. Animals

A total of 56 male Sprague Dawley rats with a bodyweight of 200–300 g were included in this study. All animals were housed according to protocol, with free access to food and water ad libitum under standardized conditions and a regular light–dark cycle for 14 days prior to the operation. All operations were performed under general anesthesia and in aseptic and sterile surgical conditions. We used an established experimental setting with subcutaneous implantation of the meshes [16]. The animals were randomly divided into eight groups according to the four meshes (PP in standard and bold structure, PP-S, and PP-B and PVDF in standard and bold structure, PV-S, and PV-B) and two time points (7 and 21 days). Each group consisted of 8 animals. The study was approved by the Institutional Review Board of the Governmental Animal Care and Use Committee (LANUV, Landesamt für Natur, Umwelt und Verbraucherschutz Nordrhein-Westfalen, Recklinghausen, Germany, protocol code 02.04.2019.A196).

### 2.2. Mesh Materials

Mesh materials used in this study were provided by FEG Textiltechnik, Aachen Germany. Mesh sizes were 2 cm^2^. The meshes were made of either PP or PVDF. Filament diameter, length, and surface were comparable between groups, while filament linking was 1.82–1.85 in meshes with standard structure and 3.25–3.33 in meshes with bold structure (see Table 1). For visualization of the mesh structures, see Figure 1. All meshes were packed under sterile conditions.

### 2.3. Surgical Procedure

After induction of anesthesia with isoflurane and analgesia (metamizole 100 mg/kg bodyweight s.c.), two meshes were implanted in subcutaneous positions in each rat. The left mesh served as negative control, and the right mesh was used for the model. Rats were weighed and placed in a supine position. The abdomen was shaved and disinfected with polyvidoneiodine solution. A median incision of approximately 2–3 cm in length was made, and the subcutaneous tissue was dissected in a blunt manner on both sides. Meshes (1 × 2 cm) were implanted in the left and right abdomen in a subcutaneous position. The right mesh was subsequently infected by instillation of 0.1 mL of a 10^6^ CFU/mL *S. aureus* (ATCC14154) suspension, as described previously [17,18]. The skin was closed with 4-0 PDS suturing material with a running suture, and anesthesia was stopped. No antibiotic treatment was given prior to or after the operation.

Animals were observed daily throughout the whole study period to assess local and systemic complications.

After 7 and 21 days, rats were sacrificed under general anesthesia with an injection of 400–800 mg of pentobarbital sodium/kg bodyweight i.p., and tissue specimens were obtained for subsequent histopathological examination. Tissue specimens for histological and immunohistochemical analysis were immediately fixed in 10% formaldehyde. 

### 2.4. Histological Assessment and Immunohistochemical Analysis

After fixation, specimens were paraffin embedded and cut into 3 µm sections. All sections were stained with hematoxylin and eosin (H&E). HE stains were evaluated by a blind observer. A histological score was calculated by adding points for inflammation, depth of inflammation, neovascularization, cellular repopulation, and foreign body giant cells in 10 high power fields (HPF) per specimen at 400× magnification, taking the average according to Cole et al. [19]. Meshes were scored at the tissue–mesh interface. Average scores for each material and time point were then compared. See Figure 2 and Figure 3 for examples of the HE stains.

For immunohistochemistry, pretreatment with Tris-EDTA (pH9) 1:100 was performed for 20 min at 96 °C. Then, tissue sections were treated with polyclonal primary antibody. For identification of macrophages, CD68 stainings were performed with a 1:100 anti-mouse antibody (Acris, Bad Nauheim, Germany). CD8 stainings were performed with a 1:100 anti-mouse antibody (Origene, Rockville, MA, USA), and CD4 stainings were performed with a 1:50 anti-mouse antibody (Origene). Afterward, ZytoChem Plus AP Polymer System was used as secondary antibody kit system.

All sections were examined with the Tissue FAXS viewer (Tissuegnostics, Wien, Austria), and analysis was performed semi-automatically and in a standardized fashion using Strataquest software (Tissuegnostics, Austria). In each slide, 5 sections with an equal diameter of 1 mm were selected at the mesh–tissue interference (see Figure 4). Percentages of positively stained cells were assessed, as visualized in Figure 5. For immunohistochemistry, only infected meshes were examined, and PP mesh in standard structure served as control group.

### 2.5. Statistical Analysis

Statistical analysis was carried out with Statistical Package for Social Sciences software (SPSS, Chicago, IL, USA) and GraphPad Prism (GraphPad Software, version 9, Boston, MA, USA). All parameters are indicated as mean with standard deviation (SD) or median with range, unless otherwise indicated. Outliers +/− 2 SD were excluded from further analysis. Differences between the study groups were analyzed using one-way ANOVA and two-way ANOVA. A *p*-value of < 0.05 was considered statistically significant.

## 3. Results

The postoperative course was normal in all animals. All animals returned to normal activity on the day of surgery. There were no overt clinical signs of wound infection during the postoperative course, and no animal showed any sign of a generalized infection. All animals survived the operative procedures and postoperative period until sacrifice.

### Histological Inflammation Score and Immunohistochemical Observations

The histological inflammation score was significantly different between the groups at 7 but not at 21 days after the mesh implantation (F(3,40) = 14.26, *p*-value < 0.0001) and (F(3,43) = 1.10, *p*-value 0.36), with the highest scores for PP-S after 7 days and PP-B after 21 days, respectively. For the results of the histological stainings and differences between the groups, please refer to Figure 2 and Figure 3. A comparison of the CD8^+^ T cells was statistically different after 21 days but not 7 days after mesh implantation (F(3,21) = 4.68, *p*-value = 0.012) and (F(3,19) = 0.47, *p*-value = 0.71). PVDF meshes in a standard and bold structure (PV-S and PV-B) showed significantly lower CD8 expressions than PP meshes in a standard structure (control group) (*p*-values 0.029 and 0.028, respectively). For examples of our staining results, please see Figure 6 and Figure 7.

The number of CD68^+^ macrophages was significantly different at 21 days (F(3,21) = 4.14, *p*-value = 0.019) but not 7 days after mesh implantation (F(3,22) = 0.07, *p*-value 0.97). PV-S meshes at 21 days showed a significantly lower expression of CD68 than standard PP-S meshes (*p* = 0.011). For representative example images please refer to Figure 8 and Figure 9.

CD4^+^ T helper cells were significantly more abundant at 7 days (F(3,22) = 4.82, *p*-value = 0.01), but not 21 days after the mesh implantation (F(3,17) = 1.65, *p*-value = 0.214). PV-B meshes showed a significantly higher CD4^+^ T cell count at 7 days than PP-B and PV-S meshes (*p*-values 0.015 and 0.03, respectively). For examples of our immunohistochemical staining results, see Figure 10 and Figure 11.

## 4. Discussion

Hernia surgery requiring mesh implantation represents an important field in abdominal surgery. The main concerns are mechanical stability and biocompatibility on the one hand, and the prevention of secondary and potentially chronic infections on the other [2]. The textile properties of synthetic meshes have previously been shown to influence bacterial adhesion [13]. Sanders et al. demonstrated in an in vitro study increased bacterial adherence to filaments with a larger diameter and to meshes with a higher mesh weight and smaller pore size [10]. However, the influence of the mesh structure with comparable total filament surface, length, and diameter on the tissue response in the presence of a bacterial infection has not been investigated.

We noticed in this study a significantly higher histological foreign body tissue reaction 7 days after implantation of standard PP meshes when compared to PVDF meshes of both structures. PP meshes in bold structure also showed a significantly lower inflammation score at 7 days compared to PP meshes in standard structure.

The tissue reaction to the mesh foreign body is mediated by inflammatory cells. CD68^+^ macrophages account for a major part of the foreign body reaction [20]. Other cell types that are commonly detected are CD8^+^ cytotoxic T lymphocytes and CD4^+^ T helper cells [21]. Macrophages reach a peak density some days after infection/implantation and the number tends to decline afterward [22]. In our model with infected meshes, we found a significant decline only for PVDF meshes in a standard structure in comparison to PP meshes in a standard structure after 21 days. For the other meshes, the number of CD68 macrophages remained stable for 21 days. Cytotoxic T cells have been described to impede wound healing [23]. In our model, we observed significantly lower CD8^+^ T cell counts at 21 days after implantation of PVDF meshes of both structures when compared to standard PP meshes. These results demonstrate the superiority of PVDF meshes in comparison to PP meshes in terms of their subsequent susceptibility to infection [11]. This reduced associated risk of infection is consistent with the fact that PVDF meshes have been applied in abdominal wall reconstruction in a prospective cohort of patients with chronic infections and enteric fistulas [12].

In the present study, standard PP-S meshes showed higher histological inflammation scores than other mesh types at 7 days, whereas no difference was noted at 21 days after implantation. This finding demonstrates an advantage of the modified mesh structure in comparison to the standard structure. A possible reason for this result may be the higher filament linking but bigger pore size. A higher pore size and thus a lower probability of fibrous bridging have previously been shown to decrease the inflammatory response [8]. However, here, the mesh material (PP versus PVDF) proved to be of greater importance than the mesh structure with respect to the risk of infection.

The present study confirms the reduced risk of bacterial adhesions of PVDF meshes in comparison to PP meshes. An important limitation of our study is that the modification of both the mesh material and mesh structure might reduce the statistical power of the analysis. Although all the implanted meshes showed a favorable clinical postoperative course and only small differences were detected, even these small differences might impact the clinical outcome significantly, given the widespread use of these materials in patients worldwide.

## 5. Conclusions

The implantation of PVDF meshes resulted in a lower risk of bacterial adherence compared to PP meshes in a contaminated implantation setting. The tissue infiltration by macrophages and cytotoxic T cells was most pronounced at 21 days in the PP group with a standard structure, while T helper cells were most abundant 7 days after implantation of PVDF meshes with a modified structure. Taken together, while the mesh material exerts a clear influence on the implant outcome, the mesh structure might still provide a minor benefit with a reduced risk of bacterial infection.

## Figures and Tables

**Figure 1 biomedicines-11-02083-f001:**
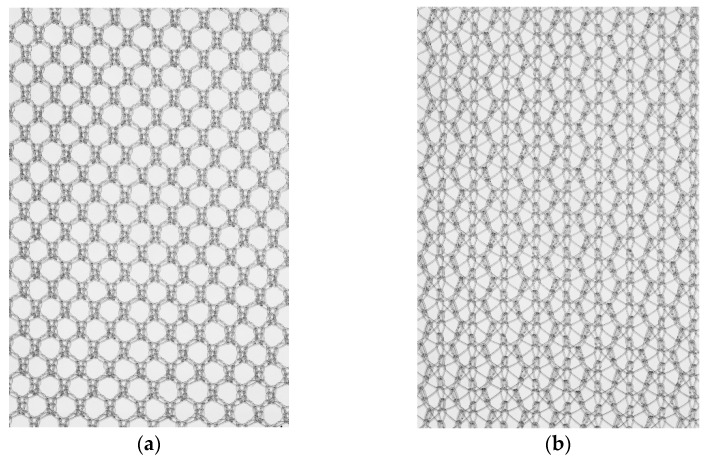
Structure of the different meshes: (**a**) represents standard structure with standard grade of filament linking; (**b**) represents bold structure with higher grades of filament linking.

**Figure 2 biomedicines-11-02083-f002:**
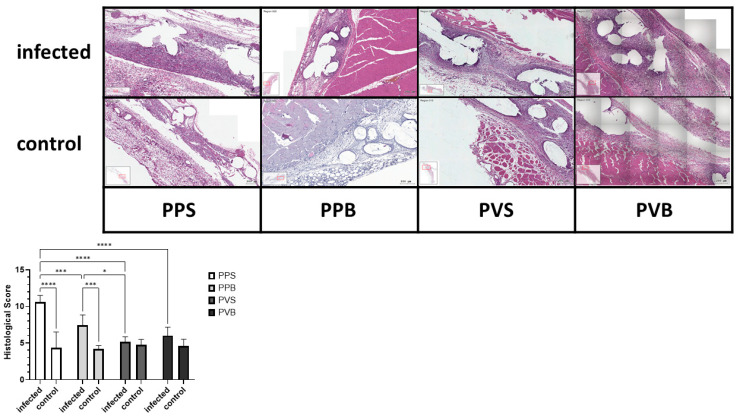
Histological score for inoculation of the different meshes with *S. aureus* after 7 days. Two-way ANOVA F (3,40) = 14.26, *p*-value < 0.0001; * = *p* < 0.05; *** = *p* < 0.001; **** = *p* < 0.0001.

**Figure 3 biomedicines-11-02083-f003:**
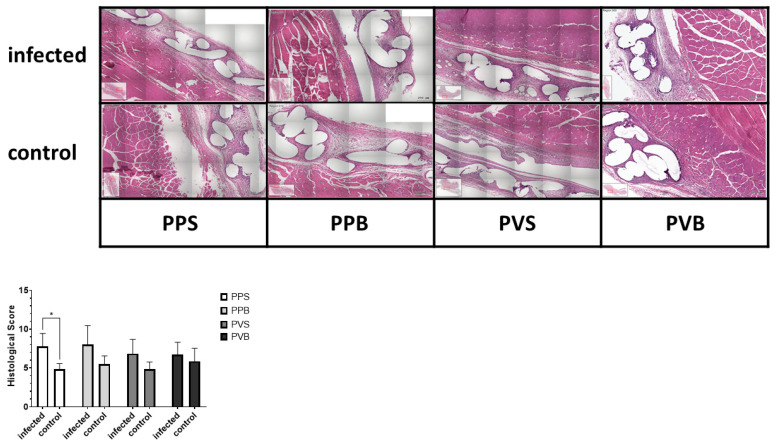
Histological score for inoculation of the different meshes with *S. aureus* after 21 days. Two-way ANOVA F(3,43) = 1.10, *p*-value = 0.36, * = *p* < 0.05.

**Figure 4 biomedicines-11-02083-f004:**
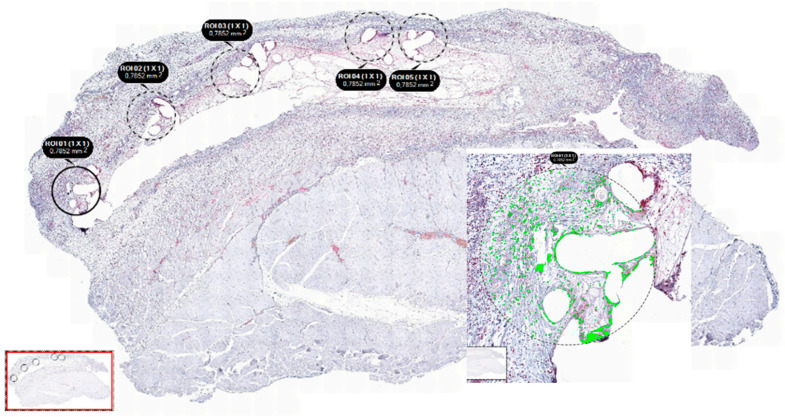
Example for manual setting of regions of interest (ROI) at the area of mesh–tissue interference and automatized counting of stained cells using the software “TissueFaxs” and “Strataquest”. Five regions of interest were set per slide.

**Figure 5 biomedicines-11-02083-f005:**
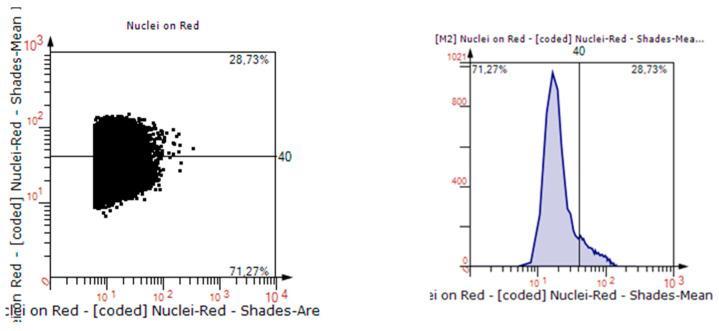
Scattergram and Histogram of the automatized counting of stained cells using the software “TissueFaxs” and “Strataquest”.

**Figure 6 biomedicines-11-02083-f006:**
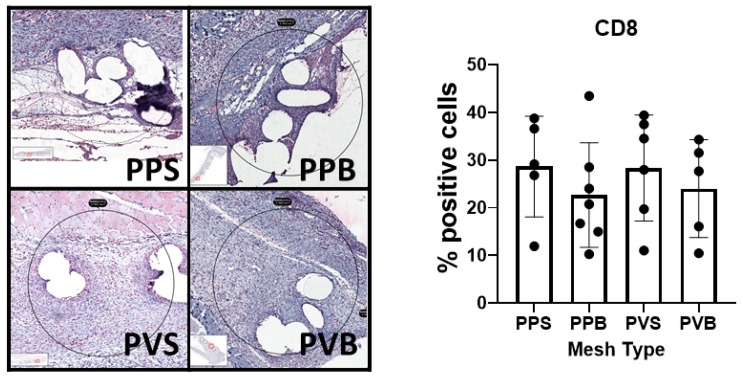
Evaluation of the expression of CD8 in the different meshes after 7 days. One-way ANOVA F(3,19) = 0.47, *p*-value = 0.71.

**Figure 7 biomedicines-11-02083-f007:**
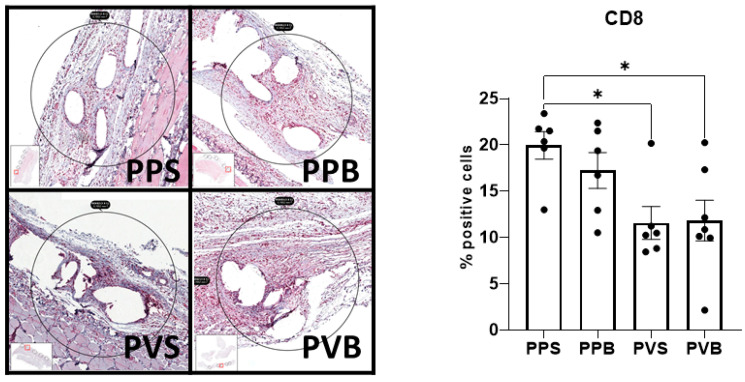
Evaluation of CD8 in the different meshes after 21 days, One-way ANOVA F(3,21) = 4.68, *p*-value = 0.012; * = *p* < 0.05.

**Figure 8 biomedicines-11-02083-f008:**
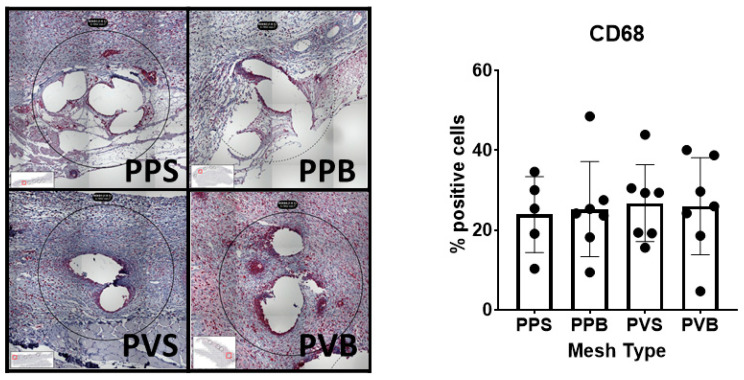
Evaluation of CD68 in the different meshes after 7 days. One-way ANOVA F(3,22) = 0.07, *p*-value 0.97.

**Figure 9 biomedicines-11-02083-f009:**
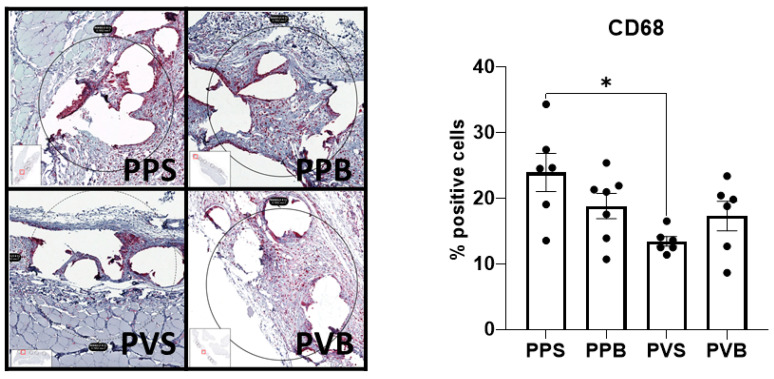
Evaluation of CD68 in the different meshes after 21 days. One-way ANOVA F(3,21) = 4.14, *p*-value = 0.019. * = *p* < 0.05.

**Figure 10 biomedicines-11-02083-f010:**
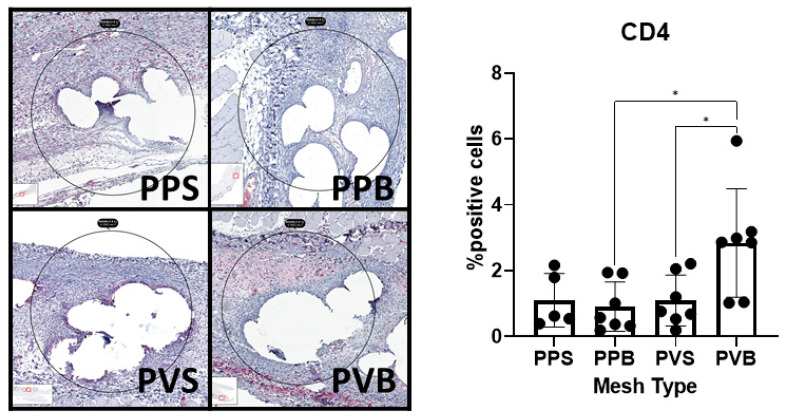
Evaluation of CD4 in the different meshes after 7 days. One-way ANOVA F(3,22) = 4.82, *p*-value = 0.01. * = *p* < 0.05.

**Figure 11 biomedicines-11-02083-f011:**
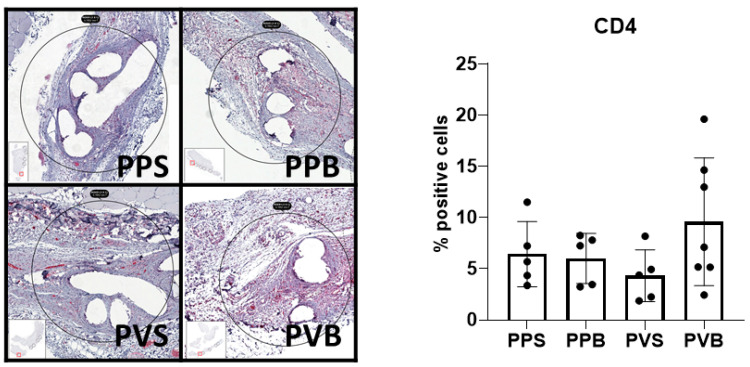
Evaluation of CD4 in the different meshes after 21 days. One-way ANOVA F(3,17) = 1.65, *p*-value 0.214.

**Table 1 biomedicines-11-02083-t001:** Characteristics of the different meshes.

	Standard Structure		Bold Structure	
Material	PP	PVDF	PP	PVDF
Filament linking	1.82	1.85	3.25	3.33
Filament diameter (µm)	165	165	165	165
Filament length (m/m^2^)	3641	3690	3692	3684
Filament surface (m^2^/m^2^)	1.85	1.87	1.85	1.86

## Data Availability

Data is available upon reasonable request from the authors.

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
