# Peer review of "Choice of Polymer, but Not Mesh Structure Variation, Reduces the Risk of Bacterial Infection with Staphylococcus aureus In Vivo"

_biomedicines, 2023, doi:10.3390/biomedicines11072083_

Round 1

Reviewer 1 Report

The paper entitled „Choice of polymer, but not mesh structure variation reduces the risk of bacterial infection with S. aureus in vivo” regards an important issue of bacterial infection occurring during surgical treatment with the use of different biomaterials. Although, the paper is in general well written and the results are important for researchers and surgeons, before publishing it require some improvement and corrections as listed below:

1.       Typo in word Germany in authors affiliation.

2.       Lack of keywords.

3.       Please unify the style in Table 1.

4.       Subsection 2.4, first sentence – should be “mm”.

5.       First sentence below fig. 2 – please correct “und”.

6.       Subsection 3.1. please check double the “( )”

7.       Subception 3.2 some error occurred with the references.

8.       Please unify the “+” sign with the CD68, CD4 and CD8 in the whole text.

9.       Figure 1 and 2 – please write the name of S. aureus correctly (italic ect).

Author Response

Dear Reviewer, 

thank you for your appreciation of our manuscript and your time taken to review it. 

We resolved all raised issues and believe that the manuscript improved notably. 

Thank you very much again

Sophia Schmitz

Reviewer 2 Report

Polymer mesh implantation is used in some abdominal surgical interventions. One major challenge is the chronic infections. The authors investigated the effect of the structure of the mesh on the infection capabilities of bacteria. Two types of meshes were used PP or PVDF. The in vivo studies evaluated the reaction by analysis of inflammatory cells. Lower CD8+ T cell were counted in 21 days after implantation for PVDF meshes compared to PP meshes. PVDF meshes was confirmed as better performing than PP meshes in respect to susceptibility to infection. Overall structure effect was minor compared to material effect. The manuscript reports novel data on the mesh infection susceptibility. Minor issues are listed below.

1.      Please cite bacteria names properly. The name “staphylococcus aureus” should be in italic format and the genus name should should be in capital letter.

2.      Some references can not be seen in the text because of format problem.

3.      Please check the figure numbering (there are two Figure 1). Also, both Figure 1 were not cited in the text.

4.      Page 5: Please correct “und”

5.      Please explain better “mean (SD) or median (range)”

6.      I suggest to explain further Figure 4 (“Scattergram and Histogram of the automatized counting of stained….”)

7.      Sections 3.1 and 3.2 can be merged under a single title.

8. Figure 5, 6, 7, 9 and 10 should be cited in the text.

Author Response

Dear Reviewer, 

thank you for your time taken to review our manuscript. We revised the whole manuscript thoroughly and corrected all mistakes.

Thank you very much again

Sophia Schmitz